# Stories of Life during the First Wave of the COVID-19 Pandemic in Italy: A Qualitative Study

**DOI:** 10.3390/ijerph18147630

**Published:** 2021-07-18

**Authors:** Silvia Caterina Maria Tomaino, Sabrina Cipolletta, Zlatina Kostova, Irina Todorova

**Affiliations:** 1Department of General Psychology, University of Padua, 35131 Padua, Italy; sabrina.cipolletta@unipd.it; 2Department of Psychiatry, University of Massachusetts Medical School, Worcester, MA 01655, USA; kostova.zlatina@gmail.com; 3Department of Applied Psychology, Bouve College of Health Sciences, Northeastern University, Boston, MA 02115, USA; i.todorova@northeastern.edu

**Keywords:** coronavirus, COVID-19, health psychology, qualitative research, personal construct theory

## Abstract

The COVID-19 pandemic has imposed on people the need to find meaning in many unprecedented ways. The aim of this qualitative study was to explore how the general Italian population dealt with government restrictions and to understand personal experiences connected with the first wave of the pandemic in light of the personal construct theory (PCT) framework. One hundred and sixteen people (over 18 years old) completed an online survey between May and June 2020. Two independent researchers ran inductive thematic content analysis on data using a specifically developed international codebook. Five major themes were identified in the participants’ narrations: difficulties, emotions, coping with lockdown measures, going back to normal, and change. The results, interpreted within the PCT transitions, showed that the pandemic represented a threat to participants’ life plans, beliefs, and certainties. Some coped with it mainly by waiting for the pandemic to end and remaining firm in their beliefs and certainties, whereas others coped by trying to find alternative ways of giving sense to this experience and reconstructing personal meanings, claiming a change in their life and in society. Differentiating personal experiences of the COVID-19 pandemic is fundamental for designing personalised strategies to promote well-being.

## 1. Introduction

The COVID-19 pandemic, as a global catastrophe with a sudden outbreak, prominently impacted social, economic, political, and personal aspects of life. A massive response from the healthcare system and general population was necessary to contain the infection. Italy was the first country after China to be significantly hit by the pandemic—by the time the WHO declared COVID-19 a pandemic in March 2020, there were 12,462 cases and 827 deaths in Italy. The Italian government imposed lockdown measures to contain the spread of the virus and support the healthcare system and workers, who were facing unprecedented difficulties and challenges at that time.

The lockdown measures imposed a halt on visiting relatives or friends, going shopping, going to the gym, going to the office, moving freely about the city, travelling, going to museums or cinemas, going to school or university, going to a bar or restaurant, and so on, thus generating significant and unprecedented disruptions in everyday life activities, habits, and certainties.

The impact this global catastrophe has had on mental health and the general well-being of the population is just starting to be fully grasped. However, the available evidence suggests that the general psychological responses seem to follow the same patterns of past viral outbreaks [1]. Symptoms such as depression, anxiety, and a general sense of unwellness have been reported in many countries since the beginning of the pandemic, especially connected to the distresses caused by the breakdown of people’s habits and routines [2].

Just two weeks after China’s COVID-19 outbreak, parts of the general Chinese population reported moderate to severe depressive and anxiety symptoms and moderate to severe stress levels [3]. Consequences of the pandemic, such as generalised anxiety disorder and depressive symptoms in the general population, were confirmed again in March 2020. In addition, poor sleep quality [4] and symptoms of post-traumatic stress disorder resulting from COVID-19 [5] were reported.

As the virus spread globally, several studies from a multitude of different countries were published and confirmed that the impact of the pandemic on mental health had similar characteristics in different populations, such as Saudi Arabian [6], Irish [7] and Spanish [8] groups.

In recent months, several studies [9,10,11] that adopted a qualitative approach examined the in-depth experience during the COVID-19 pandemic of several different groups.

Pham and Shi [11] focused on Vietnamese students studying in the USA and pinpointed eight significant causes of their distress during the pandemic: unsafe living conditions, interruption of school and work, limited access to healthcare when sick, inability to return home, isolation and movement restrictions, unstable career future, racism, and cultural factors. Another analysis [9] discovered that the pandemic has negatively impacted people with eating disorders. The situation has disrupted people’s living situation, routine and perceived control, increased social isolation, and limited access to usual support networks and healthcare services. It has also led to changes in physical activities and people’s relationship with food, including increased exposure to triggering public messages. A study involving medical workers offered insight into the thoughts and experiences of healthcare providers: the study found that the respondents perceived their job as being their “duty” to the patients, that they felt exhausted and challenged due to their new work environment, and, lastly, that they coped with the situation by improving their self-management and resilience [10].

The pandemic destroyed our everyday life certainties, making us face global, social, political, and economic disruptions, death and fear, leaving the population disoriented about the present and the future. Although overwhelming and scary, such an adverse event has provided us with challenging opportunities to find new sense and meaning in construing personal experience [12].

Currently, many studies [2,6,7,8,13] are being conducted to better understand the implications of this global emergency on the well-being of the population. However, there is still limited information on how the general population is making sense of the current pandemic and, in particular, if there are cultural differences that play a role in differentiating those perceptions. “Stories of life during a pandemic” is an international study to investigate how people are living during the COVID-19 pandemic and how they are dealing with such a situation (Blinded). The data examined in this article are part of the study mentioned above and focus on Italian citizens’ responses to point out specific contextualised meanings and lived experiences of the pandemic during the first lockdown implemented in a Western country.

The conceptual framework of this paper is the personal construct theory (PCT) developed by George Kelly [14]. PCT is an elaborate and comprehensive theory of personality that allows for a deep and coherent psychological understanding of the person experiencing life events and challenges [15].

The fundamental postulate of Kelly’s [14] theory is that “a person’s processes are psychologically channelised by the way in which he anticipates events” (p. 32). When talking about change, Kelly uses the term “movement” to precisely describe its dynamicity. According to PCT, human functioning works with the anticipation of the world, based on the awareness of replications in personal experience; this view presents the person as a scientist who constructs the world with tests and experiments that could be accepted, revised, or invalidated. Within PCT people are seen as the creators and experts of their world of meanings. Thereby, changes experienced by people are not due to external events but rather the experience of incompatibility with their usual ways of construing events, a thing that leads to the possibility of giving new meaning. The COVID-19 pandemic poses new challenges that need new meanings to be faced. PCT may offer a useful framework for understanding these changes in meaning making and helping people cope better with the situation in daily life and in therapy and, in some cases, also recover from psychological suffering and strain.

In PCT, change is defined as “transition”. Transitions are diagnostic constructs used by psychologists to describe the experience of change lived by a person and its effects on one’s construction system. Transitions have been previously used to understand the illness experience [16,17] and the present pandemic [12,18], thus providing a base for understanding the “stories of life” that the Italian respondents shared for the international study.

As PCT offers a deep and comprehensive framework to read and understand human experience, we applied it to our study to qualitatively understand behaviours, attitudes, and thoughts connected to the COVID-19 pandemic and differentiate them to design personalised strategies to promote well-being. In fact, the aim of this study was to explore how the general Italian population dealt with the restrictions imposed by the government, the lessons they have learned from such a difficult situation, and what they are most looking forward to when the pandemic is over.

## 2. Materials and Methods

### 2.1. Participants

An online survey was created by the international team of researchers and posted online on Qualtrics (https://www.storiesduringapandemic.com/ (accessed on 7 May 2020) in early May 2020 [19]. The survey took around 20 min to be completed and was available in 15 languages, which allowed for respondents from diverse countries: Bangladesh, Brazil, Bulgaria, China, Germany, India, Israel, Italy, Malaysia, the Netherlands, Puerto Rico, Romania, Switzerland, the United Kingdom, and the United States.

This paper focuses on the Italian responses. The Italian team recruited participants by disseminating the survey online, by contacting participants directly via personal and professional mailing lists and messages (WhatsApp), via specific posts on social media (Facebook, LinkedIn), daily newspapers, and by snowball sampling. The survey was available for completion from 7 May 2020 to 23 June 2020. At that time, Italy was moving into “Phase 2” of the lockdown, characterised by fewer restrictions still under semi-lockdown. During this period, funerals were allowed with a maximum of 15 participants, while religious facilities were still closed, as were restaurants, cafes, theatres, etc. Wearing a face mask and maintaining a one m distance between people was mandatory both in open and closed spaces. People were not allowed to move or travel between regions and within their region apart for those with certified work or health necessities.

A total of 116 responses were registered in the Italian sample. Two were deleted because they were incomplete; therefore, the final data set consisted of 114 full responses. Of the participants, 71.93% identified themselves as women (*n* = 82), 27.19% as men (*n* = 31) and 1 as “other”. The mean age was 36.16 years old (SD = 12.87); women had a mean age of 34.62 years (SD = 12.58), and men of 39.77 years (SD = 12.97). The marital status of the participants was single in 16.67% of cases (*n* = 19), in a relationship/engaged in 42.98% of cases (*n* = 49), married in 34.12% of cases (*n* = 39), and divorced in 2.63% of cases (*n* = 3). The highest level of education completed by participants was less than high school in 4.39% of cases (*n* = 5), high school diploma in 27.19% of cases (*n* = 31), university/post-high school education in 15.79% of cases (*n* = 18), college/university degree in 22.81% of cases (*n* = 26), master’s/post-graduate studies in 20.18% of cases (*n* = 23), and doctoral degree in 9.65% of cases (*n* = 11).

The respondents’ living situation was as follows: 7.02% lived alone (*n* = 8), 29.82% lived with their parents (*n* = 34), 21.93% lived with a partner (*n* = 25), 31.58% lived with a partner and children (*n* = 36), 2.63% lived with parent(s) and partner/children (*n* = 3), and 6.14% lived with friends/roommates (*n* = 7). In the spring months of 2020, six participants reported having been infected with COVID-19 with mild symptoms, no participants reported having been infected with severe symptoms, 57 participants reported not having been infected, and 50 participants reported not being sure if they had been infected.

### 2.2. Data Collection

The survey consisted of 20 closed-ended questions regarding demographic information and self-rated health status, as well as three open-ended questions regarding life during the pandemic; the responders were invited to answer with their own words and report their personal experiences [19].

The three open-ended questions were the following:What are the main difficulties you are facing, and how are you dealing with them?What has the pandemic taught you about what is important and meaningful to you?What are you most looking forward to after the pandemic is over, and why?

Online informed consent was presented to the respondents at the beginning of every survey, informing them of their rights and providing essential information about the study. Each participant was required to read it and express their voluntary participation in the study, recorded digitally. Inclusion criteria were being at least 18 years old and speaking one of the languages in which the survey was available. All data were recorded anonymously, participation was voluntary, and no monetary compensation was offered.

The study was approved by the Internal Review Board of the Northeastern University Internal Review Board (Protocol IRB #: 20-04-28).

### 2.3. Data Analysis

Descriptive statistical analysis was conducted on data collected through the closed-ended questions using Microsoft Excel (Excel version 2016). Thematic content analysis (TCA) [20] was conducted on data collected through the open-ended questions. The Italian research team used the international codebook, as well as country-specific definitions of themes and categories that made possible a contextualised understanding of the Italian experience. Thematic analysis following an inductive approach implies that data are coded without a pre-existing theoretical framework, thus enabling an in-depth and comprehensive understanding of the observed phenomenon; the addition of content analysis allows for the quantification and expression of the data through frequencies [20].

Following the guidelines for thematic analysis [21], the researchers first became familiar with the stories in their local language and contributed to the inductive development of the codebook; subsequently, they re-read the transcripts and started to annotate important information, then proceeded to manually code data, identifying the presence of common patterns in the codebook, defined as themes. It was decided that the coding would be done at a semantic level [21] due to the multicultural nature of the international sample. This approach relies on the identification and codification of explicit themes, intending to keep the interpretation of such themes to a minimum [22]. Finally, overreaching themes were identified and refined.

For the Italian data, two independent researchers manually coded the stories from each open-ended question to avoid possible errors in the data analysis. They then compared their results to verify agreement or discuss disagreement. If an agreement was not reached, a third researcher was consulted to grant a third independent opinion on the combined coding and to offer a solution. This process helped to increase the validity of the codification.

## 3. Results

Through TCA and coding Italian participants’ answers, multiple themes were identified; for this study’s purpose, we focus on five major themes with respective codes: difficulties, emotions, coping with lockdown measures, going back to normal, and change. Figure 1 introduces the combination and mapping of thematic codes to the PCT dimension of “transitions” through anxiety, threat, aggression, and hostility.

To contextualise and support the themes, many quotations translated from Italian will be used. To respect the participants’ anonymity, quotations will be identified by a code (P_00), age, and gender.

### 3.1. Difficulties

The main difficulties experienced by the participants related to the measures adopted to prevent the spread of the virus, such as work- and school-related disruptions, health issues, stresses, relationship problems caused by forced distance or closeness due to lockdown.

Working from home was described as an imposition causing discomfort by 32 participants, who reported feeling as if they did not have the resources and tools to adapt efficiently. Some suffered from working without human interactions and experienced many difficulties in maintaining a functional balance between their personal and professional lives when working from home or engaged in online schooling. For example, one participant reported having difficulty “keeping clear boundaries between my professional […] and personal life in a time of smart working at home, where personal and work spaces were at risk of overlapping” (P_115, F, 36).

Others wrote about feelings of burnout, experiencing a massive lack of motivation and productivity caused by the forced confinement at home, attempts to balance job and childcare, and an overall change in working conditions, such as economic difficulties, loss of income, and unemployment or fear of losing their job: “I will have to start it all over again, I will have to accept any job if I want to survive” (P_49, F, 38).

Twenty-three participants were concerned about their health and that of their loved ones; of those, some were concerned about their mental health, reporting how feeling lonely and scared was damaging to their sleep quality and to their control over personal thoughts. Three people in particular experienced the loss of a loved one due to the COVID-19 infection: “The most natural and simple gestures have now become challenges, and you have to constantly keep in mind that this illness exists and it can affect anyone” (P_60, F, 26).

Stresses connected to restrictions in mobility and protective measures were reported by 31 responders. Some reported distress from wearing masks and gloves and from making sure everything was adequately disinfected; others mentioned logistical difficulties, mainly ensuring groceries. Boredom, the perception that each day was similar to the previous, and the feeling of being “trapped” and fed up with lockdowns and harsh restrictions were also reported, such as the “inability to get far from my house for more than 2 m. Feeling trapped” (P_5, F, 31). For some, the main issue was the restriction of mobility that hindered their ability to reach their workplace or their home if they lived abroad. More generally, they felt as if their freedom was limited, and they missed out on opportunities and events. Some people reported relational problems and described living confined at home for long periods with their families as highly stressful: “This situation of health emergency pushed me to go back to live with my parents and brother in an uneasy environment” (P_76, F, 22).

On the contrary, those living far from home reported missing their families and partners. Disruptions in social life were pointed out. Many missed their friends and had difficulties with maintaining a romantic relationship or adapting to not seeing people in person; some also emphasised suffering from the absence of physical contact, which was reported as very important to feel connected: “The main difficulty I’m facing is that of being unable to be physically close to the people I love” (P_60, F, 26).

### 3.2. Emotions

Uncertainty and confusion were the feelings reported by 26 participants, both related to the immediate future and the current time. Some blamed the government for its inability, inadequacy, and delays; others reported feeling uncertain about the economic future of the country and the unpredictability of the impact on everyone’s future: “The main difficulties were many […] in particular the speed with which we switched from total freedom and underestimation of the situation to a total lockdown and into fear” (P_19, F, 25).

Worry and fear were reported by 12 participants, both related to the possibility of contracting or inadvertently spreading the virus and the fact that their lives and plans were disrupted. The participants wrote about two different fears related to the future—the fear that the pandemic would not be over soon and the fear of uncertainty: “My biggest worry is living with the fear that something invisible can be a carrier of death, and so the terror that a little distraction or physical closeness to others could be toxic” (P_11, F, 33). Sadness was reported by five participants and was mainly connected to death and suffering; for three people, it was associated with anger, as they experienced the loss of a loved one caused by the virus: “I want to riot because nobody is going to give me back my grandma” (P_52, M, 58).

Five people expressed a feeling of fatigue, mainly connected to the feeling of being unable to cope with the new situation or not being as productive as they were expecting to be, together with the perception of their job as exhausting “I am experiencing fatigue with my job, I am exhausted and very worried about when everything will be over and working will be again as it was” (P_24, F, 28). Disappointment was reported by two participants, related to the inadequacy of the government’s responses: “I feel abandoned. Alone. My mom died, and nobody tested us. They look at us, and we feel plague-ridden” (P_48, F, 29). Gratitude was expressed by only one respondent, citing “respect for those that helped others” (P_107, F, 50).

It is worth mentioning that, although not representative of the vast majority, five participants found the restrictive measures imposed to prevent the spread of the virus reassuring: “When they finally locked us at home, I felt more secure and protected” (P_82, F, 55).

### 3.3. Coping with Lockdown Measures

The respondents coped with the situation by focusing on the activities they endorsed or making sense of and adapting to the situation they were experiencing. Eight people used practical strategies such as doing things they had no time for before, creating a new routine, learning new skills, focusing on their physical health, focusing on work/study, or stopping following updates about the pandemic situation: “You pass time cooking, reorganising your closet, and realising how many things you wanted and could have done before, but never had the time to” (P_70, F, 20).

Nineteen participants, instead, coped by changing their mindset through self-reflection, trying to see what was positive in the situation or accepting and trying to adapt: “I reacted by accepting the situation, studying hard, appreciating the positive changes, for example, enjoying meals with my family, which I could not do before because I was living in another city for university” (P_80, M, 22).

Some participants wrote about using the platform Zoom or other video calling services to keep in touch with their loved ones, and others coped through being more involved with their communities or more connected with their families. One person mentioned the “balcony games”. This reference is of particular importance for Italy because, at the beginning of the pandemic, the southern part of the country was known for organising concerts and other shows on apartment balconies to show support to their neighbours: “But then the ‘balcony games’ started: they helped distract us all a little” (P_49, F, 38).

Four people did not manage to cope well, or at all, with the situation, and they admitted trying to “not think” and stay distracted, usually through endless and meaningless social media scrolling, while hoping everything would fix itself: “I try not to think and go on” (P_107, F, 50).

### 3.4. Going Back to Normal

The desire to “go back to normality” involved a strong will from the respondents to return to a state of things with precedent, both in the sense of “restarting life where they left it” and in the sense of “looking forward to gain back all the normal things they are looking forward to”.

Sixty-eight responses mentioned the will to resume normal activities after the lockdown was over. For some, that meant looking forward to being allowed to go outside in nature: “I cannot wait to go to the beach; after this reclusion, the sea will give me back my concept of vast horizons” (P_94, F, 45). Others mentioned wanting to be able to travel again or wanting to go back to their everyday life and to be able to resume their activities, especially going to restaurants, clubs, or the gym:

“The stressful situation in which I am now is making me wish for very normal things—for example, a dinner with friends or a weekend outdoors with my boyfriend—before, those things were trivial, but in such an emergency, they are not anymore” (P_42, F, 24).

Some others mentioned looking forward to having education in their life, with the desire expressed by some professors and students to resume as it was before the pandemic. This was connected mainly to the possibility of seeing their colleagues and classmates again and of going back to their university: “I would like to attend lectures in the university building, to better distinguish family life from the university one, and to take my exams in person because taking them online makes me feel uncertain” (P_77, F, 22).

The wish to go back out in society and be surrounded by other people was expressed by 24 participants, who mentioned wanting to finally gather with their friends, celebrating all the festivities missed during the lockdown. One participated reported wanting to “organise a super party with all my friends and have excessive beer and burgers—greasy, sweaty and holding each other” (P_112, M, 33).

Being reunited with partners, friends, and families from whom they were separated because of living in different cities or countries was expressed by 39 people, including by such statements as “I miss meeting [with] my brother and my friends, but also my colleagues. I miss seeing them and listening to them talking” (P_4, F, 33).

The desire to live without imposed restrictions and fear of the virus was reported by 23 respondents. This included wishing to not worry about wearing masks, using sanitising gel, and minding social distances, as well as wanting to stop living in fear of going out or coming in contact with people—as one participant put it, “being with my friends without having to pay attention to movements, distances, breaths” (P_58, M, 48). Those missing the freedom of movement expressed this desire as the ability to freely plan their decisions: “I cannot wait to be able to ‘plan’ again, whether it is a short or long trip, or a party among friends, or an experience or weekend outdoors. This pandemic did not allow us to organise and build our tomorrow” (P_71, F, 25).

Thirteen participants were not looking forward to anything; in fact, some stated that the pandemic was not an impactful event in their lives and that they did not have any difficulties in adapting to the new situation or did not learn anything. “No difficulties,” one participant said. “Following guidelines has never been a problem. Social distance and using a mask are becoming a habit” (P_51, M, 58). Another, when asked what she was looking forward to, said, “To be honest, nothing; it has been years since I dedicated this much time to my house and my loved ones. I will miss all this” (P_103, F, 46). Others claimed that nothing positive or no new learning came out of this pandemic or admitted that they already knew what was important to them. “Opposite to what I often hear, I do not think there was anything to learn from this pandemic regarding important and meaningful things […]. However, in general, I do not believe that people learned to behave in a more respectful way towards others and their environment” (P_55, M, 49)

### 3.5. Change

Participants described the changes resulting from the pandemic in their view of themselves and the world around them. Twenty-two respondents stated that during the pandemic, they have become more aware of significant social problems—these include a greater realisation that not respecting the environment leads to catastrophes, seeing more societal ills such as the fragility of the economy, and the newly found awareness of existing social inequalities. Others realised how lacking governments are in dealing with such an event as a global pandemic: “Viruses affect everyone without distinctions and without caring about borders, which is ironic and somewhat educational towards all those people who, on the contrary, build walls” (P_80, M, 22). Some participants expressed their concerns about the problems of modern society (inequalities, people’s feeling of omnipotence) and demanded change, whereas others focused more on the exploitation of nature and how they will try to actively change their behaviour to protect our planet: “That we are not absolute masters of our lives and of nature—that we need social regulation, humanity and respect for humans and nature, economy and markets—shouldn’t be the line of demarcation of humanity” (P_47, M, 58).

Some respondents reported feeling more connected with society and humanity as a whole, as if the common misfortune of the COVID-19 outbreak put everyone on the same level, reducing social differences and emphasising feelings of solidarity: “From this experience, I learnt to look around me with more careful eyes, to look at others’ fragilities with more consideration and to open myself up to listening and helping others” (P_19, F, 25).

Twenty respondents reported that they became more aware of what is essential in their life because the pandemic allowed them to spend more time on self-care and self-reflection, focusing on activities and hobbies that make them happy or revaluing their priorities: “Every once in a while, it is important to stop and reflect on what are your priorities in life” (P_15, F, 24).

The majority of these existential insights revolved around life and the meaning of existence. In fact, some people realised how short and unpredictable life is and that the pandemic is something over which they have no control; others, realising that they prioritised the wrong aspects of life, such as work over family, affirmed that relationships and experiences should be considered the real value of life: “The pandemic taught me that the truly important things are not the material ones. […] It made me realise that relationships with others are what is really important” (P_110, F, 20). Nine participants described the desire to renew the way relationships are experienced, together with a will to engage with loved ones on a deeper level. One said that she wanted to “hug people and be able to spend meaningful time together without worrying about infections” (P_95, F, 33). These participants were also re-evaluating the importance of physical closeness with loved ones: “Even though I got used quickly to staying in contact with my social circle (relatives, friends, colleagues, students) from a distance, especially after three months, I started to realise the importance of physical closeness in interpersonal relationships” (P_104, M, 41).

Five participants stated the importance of spending more quality time with their loved ones because life is unpredictable, and they could lose them at any moment. One of them said “It taught me that family and loved ones are on top and must be respected. We realise their importance only during these situations. It taught me to appreciate my health, the things I have, and to spend more time with my family” (P_54, F, 28).

An appreciation of “the small things of daily life” was reported by 52 participants: “Appreciate the small things always and be glad for them; value, rediscover, re-examine them and push for more creativity” (P_79, F, 21). A participant also said to appreciate “family, work and simplicity” (P_52, M, 58). Participants talked about comprehending things such as the pleasure of enjoying silence and an everyday slower pace, as well as the pleasure of focusing their appreciation on themselves and their achievement of self-knowledge and inner peace: “Sometimes slowing down makes you feel better and that the current society is too frenetic and engaging to the point that we never have the time to stop and think about the importance of time” (P_63, M, 43). This appreciation includes having learned to appreciate living in the moment, without taking anything for granted or wasting time on useless tasks, but instead pursuing activities that make them happy: “It taught me that nothing should be taken for granted and that everything can change in an instant” (P_86, F, 22).

Fresh air, light, flowers, and nature in general were mentioned by many as things they missed from before the pandemic and that they were learning to truly appreciate: “I may have also learned to realise the value of the scents of open air, because after many days home, even going out for 5 min on the balcony with the sun and the spring was something to treasure” (P_42, F, 24).

Twenty-seven participants reported being grateful for how privileged and lucky they realised they were in that they had their basic needs met and did not need to struggle to survive. They were overall in good health and had a family or friends beside them: “This forced stop made me realise even more how lucky I am; I have a family in good health, unity, a roof over my head and a man I love” (P_71, F, 25).

Two people also expressed the desire to remember the lessons they derived from the experience of this pandemic for the future: “I would like to write down some thoughts in a journal so that in some time, when this moment will be far away, I will be able to remember the lessons this period has gifted me” (P_19, F, 25).

## 4. Discussion

This study’s findings highlighted five major themes that characterised Italian people’s narrations during the first lockdown: the difficulties related to the pandemic, emotions, ways of coping with lockdown measures, the need to go back to normal, and change. These results need to be considered while keeping in mind that Italy was the first European country to deal with a coronavirus outbreak, which started in February 2020 [23], and that the government initially struggled with containing the virus: in a few weeks, Italy went from discovering the first few cases to a complete lockdown due to the emergence of thousands of cases [24].

This unprecedented event implied different transitions and required the implementation of different strategies and new meanings to overcome distress and uncertainty. These transitions and strategies will be discussed in light of the PCT framework.

### 4.1. Transitions in the Face of an Unprecedented Event

Respondents described the sudden change caused by the pandemic and the confusion that followed as highly stressful and scary. This experience can be described in terms of Kelly’s transition of anxiety as a situation in which an event—usually experienced as unknown or unprecedented—cannot be construed within a personal system of constructs [14]. As pandemics fortunately are not events typical of Western people’s life experience, it is not difficult to think that reacting to a virus might be beyond our usual system of constructions. Thus, the pandemic provoked anxiety, especially in its first months, when many aspects of it were unknown and difficult to construe (and still are at the time of this writing), including questions such as when will herd immunity be reached? will the vaccine work correctly? will the virus mutate? and so on. [12,18].

The results showed that participants mainly faced the pandemic as a threat. They imagined and construed its implications and possible evolutions and felt threatened, because it was not just something new and difficult to construe (anxiety) but rather something that was going to last and potentially change their lives, thus generating conflicts and doubts around personal certainties.

Threat in PCT terms describes a situation in which something profoundly important, some aspect of one’s personal identity, is challenged and put at risk. The possibility that our solid beliefs, thanks to which we construe, see, and experience the world, are in danger, not valid anymore, or need to be revised in light of what is happening around us causes profound uncertainty and discomfort. The pandemic, indeed, especially after some months, demanded that people revise their way of seeing the world, endangering their beliefs and putting them face to face with some of the scariest life experiences, such as death, natural catastrophe, global financial insecurities, and more. As seen in our results, threat was relevant to participants who experienced disruptions in school and work, had to adapt to entirely new work and learning environments, and were expected to be as productive as before or even more, given the extra time spent at home. This message was widely popular on social media, especially in the first months of the lockdown, and often led to feelings of inadequacy and distress [25].

The change from an “in-place” to a remote working environment was difficult for both students and workers due to the excessive flexibility of timetables, expectations for constant availability, and overlapping of professional and personal spaces, resulting in decreased productivity and motivation. Many participants reported a loss of income or having lost their jobs; both conditions have been proven to be detrimental to one’s mental well-being through increasing anxiety and depressive symptoms [26]. Additionally, considering that the Italian economy was in a deeper recession than that of 2009 [27], and that the unemployment rate in June 2020 was 8.8% [28], it is plausible that the resulting problems of mental health will not be resolved quickly and could even worsen if the economic situation does not begin to recover.

As shown by the literature, the implementation of lockdown measures and the excessive focus on the outbreak were reported together with worsening of an individual’s mental health conditions, resulting in an increase of anxiety, depression, and insomnia [13,29]. Similarly, continuous exposure to news about COVID-19 has been found to be detrimental to a person’s mental health, increasing levels of depression and anxiety [30] even after the strictest phase of lockdown.

As PCT focuses on constructing the human life experience as a form of movement, authors identified two different trajectories in facing this unprecedented experience: one characterised by stillness and the other by dynamism. As illustrated in Figure 2, the first may be described in terms of hostility and the second in terms of aggression.

#### 4.1.1. Hostility, or “Waiting for the Pandemic to Be over to Start Life Where We Paused It”

In PCT terms, *hostility* is defined as a constant effort to extort evidence in favour of a type of social prediction that has already been recognised as a failure [14]. Hostility was identified in the responses of some participants who explicitly wrote about COVID-19 as a nonchanging experience. This point of view is visible especially in themes regarding “going back to normality”, where many participants stated that they could not wait to have their lives back and restart from exactly where they paused them, as if the virus never existed. The strong desire to gain back the freedom that was so quickly lost, without being able and/or wanting to imagine a new way of living after the pandemic, confirms the idea of hostility as an attempt to extort evidence in favour of one’s anticipation (I believe that the COVID-19 pandemic will not change me and life will be exactly as it was before) that has proven to be invalidated (COVID-19 is changing our way of living). Furthermore, as shown by the results, many people felt trapped and helpless against the restrictions imposed on them; some even reported wanting to shut off all information they were receiving about the pandemic because it made them feel overwhelmed. Thus, in some sense, they were removing the experience around them from their perception, or even construing it as exaggerated alarmism [12].

#### 4.1.2. Aggression, or “The Lessons COVID-19 Gave Us about Ourselves and the World, and What to Do with Them”

In PCT terms, *aggression* is defined as the attempt at actively elaborating situations and events [14]. When describing their experience with COVID-19, many participants reported taking an active role and challenging themselves and their beliefs to deal with the situation and make the best out of it in practical and personal terms.

The pandemic was a time for people to reflect on many aspects of their lives, bringing them insights that forever changed the way they view and experience social phenomena. Our results indeed show that a large number of respondents wrote about gaining new awareness of global issues, as well as new insights about what is important and meaningful for them. Participants wrote about having understood or gained new awareness of what is essential in life, with relationships being seen as “the most important thing”. Many participants during the lockdown saw their relationships under a new light, giving renewed value, increasing their appreciation for the people they loved, and pushing them to seek deeper and more meaningful connections. Disruptions in relationships and the imposition of social distancing raised people’s awareness of the importance and irreplaceability of meaningful face-to-face relationships and physical contact with loved ones. Despite this, many people dealt with the separation during the lockdown using other resources to keep in contact with loved ones, such as video calls. Even though this way of communicating was perceived as “depersonalising”, it was also considered a good way to keep in contact and reduce stress. Our results confirmed the findings in the literature about online social support and emotional sharing being associated with positive effects on mental health and well-being [31] and about social connectedness having benefits in relieving stress and fatigue [32]. As social resources could be scarce during the period of the pandemic, psychological support via digital media together with telemedicine or peer support groups could be an important safeguard, especially for the vulnerable population, such as people with pre-existing psychiatric disorders, older, and/or marginalised people [33].

These results also align with a study that highlighted how individuals give new meaning to mundane activities [34], and, as predicted by Buheji and Ahmed [35], many also discovered their inner strengths. As Yang and colleagues [36] pointed out, the COVID-19 pandemic improved the Chinese population’s tendency to construct meaning from negative experiences; being able to do so was connected to increased resilience, together with positive adjustment and diminished distress. *Aggression* was interpreted especially from the stories of those participants who saw this situation as an opportunity for change [12]; thus, said participants dynamically tried to find personal balance between energies, emotions, everyday pace, and activities in dealing with the lockdown, restriction measures, and difficulties experienced, allowing themselves to feel both pain and hopefulness, reaching out for help and support when needed.

### 4.2. Limitations

The study’s results must be contextualised in the first period of the pandemic, strictly connected to the personal experience of Italian participants. It is worth noting that participants were spread out over the whole national territory and were affected differently, especially in the first phase of the pandemic. Thus, the personal stories reported are not representative of the general population. Furthermore, participants’ recruitment was via direct contacts and snowball sampling, potentially involving mainly people who were part of the same social group and limiting the possibility of involving people from different groups.

## 5. Conclusions

The present study offered a narrative insight into how the Italian population has made sense of the COVID-19 pandemic and its effects. What has been observed could be used to better contextualise the already-existing quantitative research on the topic. Because qualitative research methods are utilised for in-depth analysis of a topic, they play a significant role in further explaining and uncovering the underlying links between events that are not detected by quantitative research, suggesting dynamic processes, developments, and associations that are often overlooked [37].

As was recently pointed out, such a complex and unprecedented period as the present provides many opportunities for changes in personal and collective constructions [12,18]. The pandemic, in fact, as an experience that has threatened our solid certainties and beliefs, can be viewed as a unique opportunity to claim a change in a more prospective way, giving us the chance to revise our vision of the world and make it more relevant.

For some people, the pandemic represented an insurmountable negative experience that generated so much fear and uncertainty that it led them to try to go back to normality—especially to life as it was—despite the evidence that the pandemic made this return impossible. This attempt to validate a prediction that already proved to be a failure determines an arrest in movement and the impossibility of seeing or making changes. For participants who experienced it so, COVID-19 was described as something offering nothing to learn and generating no change in their perspectives and beliefs.

On the contrary, for many, the fears and questions pointed out by the pandemic fostered a reconstruction of existential beliefs and values. This proactive reconstruction of personal meaning helped people to make new experiences and create new meaning in their lives and tell a shared story in which the lesson is that “having meaning in life makes life worth living” [38].

## Figures and Tables

**Figure 1 ijerph-18-07630-f001:**
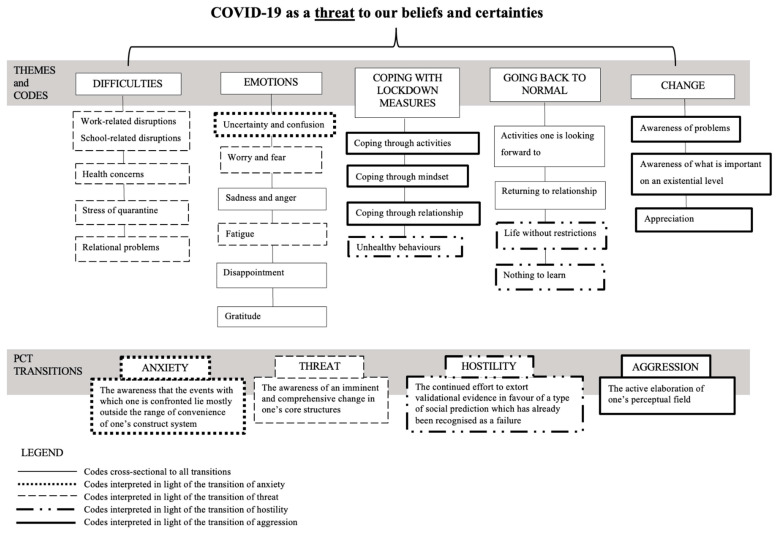
Map of themes and codes together with the PCT transitions and combinations of codes.

**Figure 2 ijerph-18-07630-f002:**
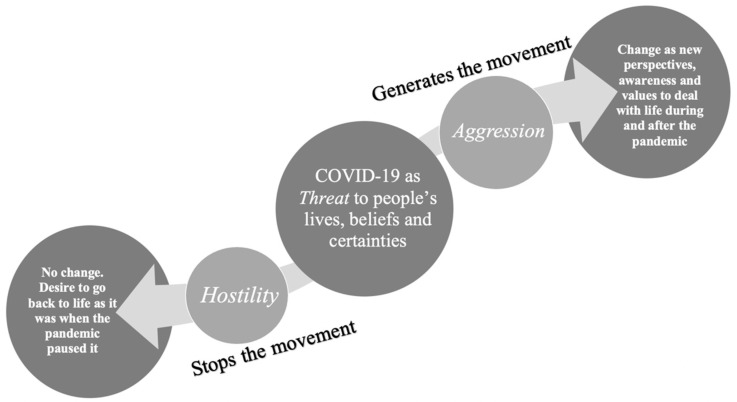
A dynamic representation of different ways to cope with the threat represented by the COVID-19 pandemic.

## Data Availability

Considering the study uses narrative data, data will not be made available due to ethical and privacy restrictions.

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
