# Peer review of "Stories of Life during the First Wave of the COVID-19 Pandemic in Italy: A Qualitative Study"

_ijerph, 2021, doi:10.3390/ijerph18147630_

Round 1

Reviewer 1 Report

This version is an improvement. There is less danger of the paper being misused now that aggression and hostility are no longer in the Abstract and with the other revisions with respect to these terms.

Yet, as per my last comment, “Notwithstanding the problems with use of words hostility and aggression, for the current manuscript to be suitable for publication, there would need to be a good chance that this personality, personal-construct-theory-based psychotherapy would substantially benefit lots of people in the general population. I do not think this is the case.”

In other words, the authors still haven’t included any information or evidence suggesting that there’s therapeutic benefit or relevance to this personality, personal-construct-theory-based psychotherapy which forms the subject of their paper.

For these reasons and because I still believe there is potential for misuse of the article, I don’t think the manuscript is suitable for publication in a public health journal.

Author Response

As suggested, the authors agreed on the importance of stating more explicitly the benefits of adopting a Personal Construct Theory approach in the Introduction Section, as following: “Within the PCT people are seen as the creators and experts of their world of meanings. Thereby, changes experienced by people are not due to external events but rather by the experience of incompatibility with their usual ways of construing events, a thing that leads to the possibility of giving new meaning. COVID-19 pandemic is posing new challenges that need new meanings to be faced. PCT may offer a useful framework for understanding these changes in meaning-making and helping people coping better with the situation in daily life and in therapy, and, in some cases, also recovering from psychological suffering and strain.”

Furthermore, all authors believe that our work's strength point focuses on how people used to give meaning to their personal experience during the first wave of the pandemic. The PCT approach focuses on the meaning that people give to their experience; thus, our work explores what possible construction of meaning was experienced and acted by Italian people living that situation. Exploring these is of great importance for human sciences because practitioners should use that bottom-up information during and after the pandemic to help people coping better with the situation and, in some cases, also recovering from psychological suffering and strain.

Reviewer 2 Report

Excellent research!

Author Response

Thank you for reviewing our work and for your positive feedback, it is very important for all of us.

Reviewer 3 Report

I would highly support this paper since it clearly describes the meaning of the pandemic to people, not some statistical numbers but real emotionas and reactions of the people who went through that stage of pandemic. This information is oustandingly important for common people, for scientists and practinioners since after a year pandemic is still there and more information is needed to cope with it and recover.

Author Response

Thank you for reviewing our work and for this positive feedback. We are very pleased to hear that our work is recognized as an important first step to move in the direction of focusing on people and their recovery during and after the present pandemic.

This manuscript is a resubmission of an earlier submission. The following is a list of the peer review reports and author responses from that submission.

Round 1

Reviewer 1 Report

Thank you for the opportunity to review this qualitative study. I have the following major and minor comments for the authors to consider.

Specific comments:

  1. "An online survey was created by the international team of researchers and posted online in early May 2020" - the authors need to provide more detail on how the survey link was disseminated and what platform was the survey conducted over?
  2. Given the use of snowball sampling, how did the authors guard against duplicate responses? E.g. IP filtering?
  3. How was the survey developed? Were any pilot studies performed?
  4. Please change "regime of semi-lockdown" to "under semi-lockdown". Would be useful to elaborate on these restrictions.
  5. On average, how long did it take to complete the survey?
  6. "... stories in their local language" - I thought only responses from Italy were analyzed in the current study?
  7. Figure 1 is slightly messy, suggest to tighten the labeling. Also, how did the authors decide on the pairing of the emotions, e.g. "guilt and fatigue"? The word "legend" was misspelled as well.
  8. As mentioned by previous authors, in the case of COVID-19, people may have learned to live with “less,” and they may have found more time for leisure activities like quiet reading, honing a new skill or craft, or simply having a meal as a family (citation: pubmed.ncbi.nlm.nih.gov/32943541). Beyond grief and depression, self-reported positive psychological changes, or posttraumatic growth is possible, albeit people simultaneously struggled with mental illness. A longitudinal study is necessary to ascertain this of course.
  9. As resources could be particularly scarce during a serious pandemic situation, timely psychological support could also take many forms, including telemedicine and informal support groups (citation: pubmed.ncbi.nlm.nih.gov/32380875). This should be mentioned.
  10. In a recent study, depression and anxiety were also found to be prevalent even during the post-movement lockdown. This should be articulated in the discussion.
  11. Several other study limitations exist. The lack of validated scales and benchmark indices on psychopathology, and likely selection bias (of unknown extent because attrition cannot be evaluated) are highly problematic and should be mentioned.
  12. "This phenomenon can result in excessive positivity (I can do everything and even more) or negativity (I won’t be able to do anything)" - verbatims should be enclosed by a pair of quotation marks.
  13. The conclusion brings out a lot of new points rather than provide a succinct and cohesive summary for the paper. Please rewrite and shorten the conclusion paragraphs.
  14. Please include a data availability statement.

Reviewer 2 Report

The authors’ use of the terms hostility and aggression is unacceptable in a public health publication and has great potential to create unintended consequences.

The authors indicate that “[t]he conceptual framework of this paper is the personal construct theory (PCT) developed by George Kelly. The PCT is an elaborate and comprehensive theory of personality that allows for a deep and coherent psychological understanding of the person experiencing life events and challenges.”

“In PCT terms, aggression is defined as the attempt at actively elaborating situations and events.”

“In PCT terms, hostility is defined as a constant effort to extort evidence in favour of a type of social prediction that has already been recognised as a failure.”

Even so, to everyone outside of personality psychology/psychiatry, the terms mean the following:

Aggression, Merriam-Webster’s Collegiate Dictionary (11th ed. 2012) means (1) “a forceful action or procedure (such as an unprovoked attack) especially when intended to dominate or master” or (2) “the practice of making attacks or encroachments especially unprovoked violation by one country of the territorial integrity of another,” or (3) “hostile, injurious, or destructive behavior or outlook especially when caused by frustration”

Hostility, Merriam-Webster’s Collegiate Dictionary (11th ed. 2012) means (1b2) “hostilities plural : overt acts of warfare”

Why is this relevant? Because there is an excellent chance that this research will be misused. That is, it will be used by governments, public mental health leaders, and psychoanalytic/psychodynamic professionals to drum up support for these services by invoking fears of violence.

That was what happened with

Jianbo Lai, Simeng Ma, Ying Wang, Zhongxiang Cai, Jianbo Hu, Ning Wei, Jiang Wu, Hui Du, Tingting Chen, Ruiting Li et al., Factors Associated with Mental Health Outcomes Among Health Care Workers Exposed to Coronavirus Disease, 3 [J]ama Network Online e2039676 (2019), which was one of the articles cited by

Kelly Posner Gerstenhaber & Keita Franklin, Congress Must Pass the Dr. Lorna Breen Health Care Provider Protection Act, The Hill (Jan. 6, 2021 8:00 AM EST), https://thehill.com/blogs/congress-blog/healthcare/532835-congress-must-pass-the-dr-lorna-breen-health-care-provider  The article stated that “[a] recent study in the Journal of the American Medical Association found that among some 1,250 medical professionals working with COVID-19 patients in China, more than 50 percent reported symptoms of depression, nearly 45 percent noted symptoms of anxiety, and over seven in 10 reported distress.” The author was a Columbia University Medical Center suicide researcher who benefitted tremendously from passage of the Dr. Lorna Breen Health Care Provider Protection Act (its provisions ended up getting adopted into the American Rescue Plan Act of 2021, signed into law on March 11, 2021), which grants academic medical centers up to $140 million, the opportunity to conduct psychiatric medical research on health professional employees and an “education and awareness campaign directed at health care professionals,” “encouraging health care professionals to seek support and treatment for their own mental health and substance use concerns” and “help such professionals to identify risk factors in themselves and others and respond to such risks.”

This suicide research has extremely limited potential for benefit. Passage of the provisions will allow unacceptable violations of physician-employee privacy (in violation of ADA prohibitions on medical inquiries), and it detracted attention from very important structural reforms.

Notwithstanding the problems with use of words hostility and aggression, for the current manuscript to be suitable for publication, there would need to be a good chance that this personality, personal-construct-theory-based psychotherapy would substantially benefit lots of people in the general population. I do not think this is the case.

While this article might be suitable for a personality, psychodynamic therapy, or clinical psychiatry/psychology journal, I do not believe it is suitable for publication in a public health journal, even if the authors were able to clarify and avoid problems with the terms hostility and aggression is both the abstract and the body of the manuscript.